# Machine-Learning-Enhanced Procedural Modeling for 4D Historical Cities Reconstruction

Beatrice Vaienti *, Rémi Petitpierre, Isabella di Lenardo and Frédéric Kaplan

Digital Humanities Institute, EPFL, Swiss Federal Institute of Technology in Lausanne,
1015 Lausanne, Switzerland; remi.petitpierre@epfl.ch (R.P.); isabella.dilenardo@epfl.ch (I.d.L.);
frederic.kaplan@epfl.ch (F.K.)
* Correspondence: beatrice.vaienti@epfl.ch

**Abstract:** The generation of 3D models depicting cities in the past holds great potential for documentation and educational purposes. However, it is often hindered by incomplete historical data and the specialized expertise required. To address these challenges, we propose a framework for historical city reconstruction. By integrating procedural modeling techniques and machine learning models within a Geographic Information System (GIS) framework, our pipeline allows for effective management of spatial data and the generation of detailed 3D models. We developed an open-source Python module that fills gaps in 2D GIS datasets and directly generates 3D models up to LOD 2.1 from GIS files. The use of the CityJSON format ensures interoperability and accommodates the specific needs of historical models. A practical case study using footprints of the Old City of Jerusalem between 1840 and 1940 demonstrates the creation, completion, and 3D representation of the dataset, highlighting the versatility and effectiveness of our approach. This research contributes to the accessibility and accuracy of historical city models, providing tools for the generation of informative 3D models. By incorporating machine learning models and maintaining the dynamic nature of the models, we ensure the possibility of supporting ongoing updates and refinement based on newly acquired data. Our procedural modeling methodology offers a streamlined and open-source solution for historical city reconstruction, eliminating the need for additional software and increasing the usability and practicality of the process.

**Keywords:** procedural modeling; 4D urban reconstruction; 4D city modeling; GIS; HBIM; historical maps; machine learning; 3D building modeling; CityJSON; vectorization

## 1. Introduction

The generation of 3D models depicting cities in the past holds great potential for documentation and educational purposes. On the one hand, they can be used to improve heritage accessibility for the general public, profiling themselves as spatial and temporal aggregators, and showing documents and information that would otherwise be difficult to explore [1]. On the other hand, these models can be seen as the scholarly outcome of a process of historical spatial synthesis, introducing a new form of academic publication for researchers [2]. Moreover, their creation could open new avenues for spatial quantitative analyses such as population estimations [3] and visibility assessments [4]. Similar use cases are diffused in contemporary urban 3D representations [5]. Despite evident benefits, the creation of such models is often hindered by the substantial time investment and specialized expertise they demand, as well as by specific challenges related to working with historical data.

This paper aims to propose a set of tools, in the form of an open-source Python library, to help practitioners in generating informative and scientific 3D or 4D urban models in an easy and documented way. The main challenges related to this objective will be presented and addressed point-by-point along with existing solutions and the proposed methodology.

**Data Incompleteness.** Firstly, it is crucial to acknowledge that historical information inherently tends to be incomplete by default. Historical data are usually fragmented, missing, or conflicting, compelling researchers to amalgamate and interpret multiple sources, while occasionally resorting to educated guesses to fill in the gaps.

In fact, when aiming at creating a 3D representation of a building, to ensure the creation of an accurate spatial representation, it is imperative to possess all the necessary data. For instance, without the inclusion of building height information, it would be impossible to generate an extrusion of the footprint and accurately depict the 3D structure. This process must be executed in a manner that preserves the informative value of the model and ensures transparency when guesses are operated. Meanwhile, at an architectural scale, we may resort to educated guesses; one way to solve this contradictory objective in large-scale scenarios is to use statistical machine learning methods to infer the most probable values in an objective way, based on available data. While deep learning techniques are well suited to process remote sensing data [6,7], already vectorized GIS datasets represent an easier case that can effectively be tackled by classical machine learning algorithms. Within this category, we favor the use of ensemble methods over regression or nearest neighbor algorithms, because of their ability to deal with missing values in input parameters.

The concept has been investigated by various studies that concentrate on urban reconstruction and the modeling of 3D cities. Biljecki et al. [8] analyzed how to fill their dataset with height information by predicting the number of floors, and selected Random Forest as the best algorithm. They then used the completed information to construct a Level of Detail 1 (LOD1) 3D model of the city, i.e., a spatial extrusion of the footprints. The prediction of the number of floors was also addressed by Roy et al. [9], focusing on residential buildings and selecting instead the Gradient Boosting algorithms. Similarly, other examples focused on the prediction of roof types [10], type of building [11], constructive technique [12], and year of construction [13,14]. On the point of view of historical applications, Farella et al. [15] presented a methodology for inferring missing height information in 2D datasets obtained from historical maps, considering the performances of various machine learning models and using the obtained information to create a LOD1 3D model. These machine learning approaches proved to be effective in enhancing the completeness of the resulting model, providing a more comprehensive representation of the urban environment.

**Cultural specificity.** One of the main challenges of historical city reconstruction is that every case study is unique. Not only is the architecture changing in every culture and every region, but the available data contributing to the remodeling can be very disparate. The solution to this challenge is two-fold. In order to achieve generality, an approach must prioritize easy customization and flexibility. Our proposed framework, founded on procedural modeling, offers complete parameterization, allowing for seamless adaptation and extension to address a wide range of modeling challenges. We also provide strategies to deal with parameters that might be totally unavailable for a specific city. Second, any purposely generic solution should be open-source. Indeed, the unique nature of cities and the multiplicity of research project objectives compels technical solutions to constant adaptation and evolution. In this regard, we consider closed commercial solutions to be a dead end, as no tool is anywhere close to comprehensiveness. The ability to alter and adapt the code to the need of multiple projects and approaches is fundamental.

**Iterative nature of scientific projects.** When working on historical data, we may not only be confronted with the initial problem of incompleteness, but we might also consider the possibility for additional information to become available in the future. Indeed, the collection of urban historical data is generally iterative, and the ability to incorporate new data into the system, and thus dynamicity, is therefore absolutely essential.

We achieve a fully dynamic model by designing a system where geometry is generated based on footprints and documented parameters. In this approach, the partial availability of data and the associated uncertainty no longer constitutes an obstacle to the creation of an informative model. Instead, the model remains editable without compromising any previously completed operations, as the geometry is treated as a temporary representation

of the available information. Furthermore, we voluntarily favor the use of relatively frugal machine learning models to predict missing data. This allows us to retrain the machine learning models whenever new information becomes available, leading to real-time estimates, and a scalable resource-efficient framework. The ability to continuously update and refine the machine learning models based on newly acquired data ensures that our historical representations remain up to date and reflective of the most accurate information available, while continuously documenting the process used to infer missing values. We value the use of interpretable machine learning algorithms to ensure traceability. Incidentally, such models can provide new insights on the studied city by handing a data-driven dependency scheme across parameters.

**Subjectivity of the reconstruction and interpretation.** Recent research studies in cartography often consider the map as an artifact culturally constraint, rather than a mere projection of the territory [16]. The process of mapping itself is made of arbitrary choices, based on cultural factors and intentional decisions. Maps and other sources can thus offer a different or even contradictory description of the city.

For this reason, it is essential to document the metadata associated with the reconstructed models (i.e., all the attributes that describe the modeled object and the respective sources) and maintain a record of the decisions and hypotheses that were assumed by researchers during the (automatic or manual) reconstructive process. This concept is also addressed as *paradata* [17,18]. Maintaining this methodology is of utmost importance to ensure transparency and scientific rigor, as well as to enable future users to understand the sources used, the underlying assumptions, and the decision-making process involved in the modeling process [2,19,20].

This need of comprehensive documentation calls for the use of a well-structured format to encode the 3D urban representation. In this context, 3D GIS shows the capability to support the informative nature of the model and facilitate spatial approaches [21]. Among the standards that are available to encode 3D GIS datasets, CityGML [22] emerges as a prominent XML-based format for the geometric and semantic representation of cities. However, while CityGML offers great power and functionality, its adoption can be challenging. To address this concern, we turn our attention to CityJSON [23], a compact and developer-friendly JSON encoding of CityGML. CityJSON serves as a compelling alternative for generating semantic and georeferenced 3D city models while maintaining interoperability with CityGML. An advantage of CityJSON relies in its support for the easy creation of extensions to the core data model. Such modularity proves particularly beneficial for addressing the specific needs of historical models, as researchers may have different requirements for data representation based on their unique study cases. To address the need for effective communication regarding the level of detail in 3D city models and their adherence to reality, Biljecki et al. [24] proposed an improved system to classify the Levels Of Detail (LODs) based on the taxonomy developed for CityGML. This system introduces 16 LODs, which refine the original five main categories by adding four subgroups (0.0, 0.1, 0.2, 0.3) to each original category (1, 2, 3, 4). In our project, we adopt this improved system, and specifically, we refer to our LOD2 modeling as LOD2.1, as it includes roof overhang and additional details beyond the LOD2.0 level of modeling.

In our research, we aimed to adapt the CityJSON format to our needs by incorporating the necessary fields to accommodate all the parameters needed for the procedural modeling scripts. Furthermore, we included fields to track the provenance of the parameters, ensuring transparency and traceability. To achieve this, we developed our own CityJSON extension, referred to as Historical CityJSON [25]. This extension underwent further refinement to align the attributes with the needs of the previously described LOD 2.1 level of modeling, which offers a higher level of detail in the representation of city elements.

By using this structure as the basis for the procedural modeling process, we can ensure both structured data and dynamism. The combination of GIS and procedural modeling has already been used in cases dealing with historical representations [26,27], and for contemporary representations [28–30]. Moreover, the availability of open-source

and commercial tools for large-scale procedural modeling, such as, respectively, the BCGA Blender Addon and ArcGIS CityEngine, has significantly contributed to the accessibility of GIS-based 3D reconstructions. These tools contribute to making the methodologies more accessible to a broader user base. However, it is worth noting that even with open-source tools, like the BCGA Blender Addon, users may need to split their pipeline and generate the spatial geometry within a dedicated 3D modeling software. This fragmentation can add complexity to the workflow, and requires users to possess a certain level of expertise in multiple tools.

With these goals in mind, and also with the objective of avoiding reliance on proprietary software or requiring users to navigate through 3D modeling applications, we developed a set of tools in the form of an open-source Python library. This toolset addresses two main steps: (1) filling the gaps in 2D GIS datasets by letting the user choose which fields he/she is interested in filling, and which fields should be employed as predictors for the inference; (2) a set of procedural modeling functions that make it possible to transform the completed 2D geodata in a 3D CityJSON containing multiple LODs. As highlighted by Biljecki et al. [24,31], there is not an intrinsic higher value in higher LOD models, since we may prefer lower LODs for a specific quantitative analysis, hence the interest in keeping all the three available LODs (LOD0, LOD1.0, LOD2.1) encoded in the model. This article aims to provide an in-depth exploration of the library's features and present various use cases to illustrate its practical applications. Our proposal offers a streamlined approach, allowing for the direct generation of 3D LOD2.1 models (including modeled roofs) from GIS files, and eliminating the need for additional software such as Blender.

Our pipeline can be initiated from any cartographic source. However, in order to showcase the effectiveness of our framework, we present a practical case study. Specifically, we use a set of footprints of the Old City of Jerusalem from 1840 to 1940, which was primarily extracted from a series of 10 maps. Footprints were first recovered using semantic segmentation, then translated into vector geometries. In order to be usable for 3D modeling, automatically extracted vector data were cleaned and simplified, using a dedicated algorithm to keep only the main orientations, while keeping sharp corners and avoiding the appearance of slivers between buildings. The data were then enriched with secondary sources. The result of this process is a GIS dataset similar to many other historical datasets. We show how our library is able to seamlessly complete the dataset, and generates its 3D representation using CityJSON as a format capable of retaining the hierarchical structure of information. Four typologies of roofs are supported, and were determined on the basis of the needs of the case study: flat roof, lowered dome, hip roof, gable roof. We release these scripts as an open-source library, which opens the possibility for the community to integrate them with new typologies.

## 2. Methodology and Approach

In this section, we will present the set of tools that were developed, following a step-by-step approach. We also demonstrate the actions performed on our dataset. In fact, although our library has been designed to be usable with any geodata, in Section 2.1, we will describe the features of our dataset and the process that we employed to create it. In Section 2.2, we will then present the parameters that are employed by the procedural modeling scripts, and in Section 2.3, we will describe our strategy to overcome data incompleteness. This step is crucial to ensure that all the necessary parameters for procedural modeling are available and accurately documented. Among the strategies, we propose a machine learning approach to fill gaps in fields that are partially present in the dataset: in Section 2.4, we will delve into this first set of tools. Our tools offer the advantage of being generic, allowing the user to choose the target fields and their related predictors, thus enabling greater customization in the prediction process. We will present the techniques employed to infer missing information, and discuss the significance and performances of this statistical data completion process. Being aware of the strong dependency of these

quantitative results with the available data, we will compare the performances of the algorithms on our dataset, avoiding quantitative comparisons with different case studies.

Finally, in Section 2.5, we will present the operations applied to transform a GIS dataset into a 3D CityJSON file using our library. We will provide a detailed description of the algorithms that we implemented for the generation of hip and gable roofs.

### 2.1. From Cartographic Sources to a GIS Dataset

To create the vector data, we employ semantic segmentation to automatically extract building footprints from 10 historical maps, published between 1838 and 1947. Then, we vectorize the footprints as high-quality vector data, by designing an algorithm especially adapted for 3D modeling. We combine the 10 maps in a diachronic vector dataset by detecting the first and last appearance of the polygons. To further enhance the database, we incorporate additional details, such as the number of floors, construction materials, and roof types, from a secondary source [32].

In our case study, the 10 maps are first georeferenced manually using a GIS software. The second step is to semantically segment the Jerusalem maps. Our primary objective is to obtain a diachronic dataset that captures the temporal information regarding the initial and final occurrences of building footprints. Since the latest maps exhibit higher cartographic precision, we use them as references for the vector footprints, and compare them with the other vectorized maps. This allows us to detect the appearance of buildings in time while using the best available quality for the footprints.

In particular, the building footprints are obtained from two specific maps: one depicting the Old City in detail (scale 1:2500, from 1947) [33], and one depicting its surroundings (scale 1:5000, from 1938) [34]. Indeed, solely relying on the 1:5000 map would be insufficient: this 1938 map from the survey of Palestine, divided into four sheets, presents excellent-quality descriptions of the area outside the walls, but lacks detail inside them. The revised map from 1947, realized at a scale of 1:2500, focuses on the Old City with greater detail, which makes it possible to also segment adjacent buildings.

The semantic segmentation aims to extract the footprints or built-up areas by recognizing the contours on the historical maps. To proceed with the training for the semantic segmentation, a total of 252 patches (each measuring $1000 \times 1000$ pixels) are manually annotated from the 10 maps, and we combine them with 132 patches selected from the Historical City Maps Semantic Segmentation Dataset [35]. Annotations are performed following the guidelines prescribed by Petitpierre [36], and a simplified version of the ontology they propose. The annotation classes consist firstly of the contours of built features (i.e., footprints), and secondly of the built-up areas themselves. This methodology makes it possible to retrieve precise building instances, even in the case of adjacency. A separate convolutional neural network is trained for each of these two tasks, following the same procedure as Petitpierre [37]. We use the dhSegment framework [38] and a simple UNet architecture [39], with a ResNet101 encoder [40].

This step creates two binary masks: the first corresponding to building contours, the second to built-up areas. The second mask is useful, for example, for discarding closed geometries that do not correspond to built-up areas. These could be courtyards, for instance, or other geographical objects, e.g., squares bounded by perimeter walls.

The step of vectorizing 2D geometries is a demanding part of the 3D model creation process. On the one hand, geometries must be simplified to avoid the aliasing effect inherited from the raster output of the neural network. Secondly, it must avoid the pitfalls of existing vectorization algorithms, such as OpenCV and scikit-image's contour functions, or QGIS's raster-to-vector module. These out-of-the-box tools often compute geometries independently for each polygon. This leads to inconsistent results, such as aliasing, neighboring polygons not sharing edges, and undesirable slivers and overlaps. On the contrary, desirable qualities of a vectorization algorithm designed for 3D data generation would include keeping sharp building corners and the ability to parameterize the level of simplification of the vertices, without affecting the local coherence.

As the vectorization algorithm comprises a relatively large number of steps, we will present it in a sequential form, which appears clearer. The main steps are illustrated in Figure 1. The corresponding Python code is released along with this article.

1. *Thinning.* First, a thinning or skeletonization algorithm is used to obtain single-pixel-wide contours [41].
2. *Identification of the connected components.* The identification of the connected components makes it possible to assign an index to each closed geometry.
3. *Removal of non-delimiting lines.* Lines surrounded by the same connected component on both sides are non-delimiting. They constitute noise in the sense that they do not allow the demarcation of distinct building instances. Based on this criterion, they are automatically removed.
4. *Corners detection.* The Harris Corner Detector is used to precisely locate prominent building corners [42]. It is supplemented by OpenCV's cornerSubPix function, which refines the position to subpixel accuracy.
5. *Vectorization.* The vectorization algorithm treats the map as a network. First, it detects all the nodes in the thinned image (i.e., intersections and corners detected at the previous step). Second, it follows the edge paths to reconstruct the segment paths. At this stage, the geometry of each segment is simplified using the Douglas–Peucker algorithm [43]. The level of simplification is controlled by the parameter $\epsilon$.
6. *Removing duplicated segments.* This step checks that detected segments are not duplicated.
7. *Reconstitution of the polygons.* To reconstruct the polygons, the nodes directly adjacent to each connected component are retrieved. The segments are adjacent to the polygon if both their starting and their terminal nodes are adjacent to the connected component. Then, simply following the adjacent segments one after the other makes it possible to reconstruct the cycle of each polygon.
8. *Polygon hierarchy and orientation.* Polygons can encompass "donut holes", which should be oriented counterclockwise, according to the shapefile format convention, while the outer cycle should be oriented clockwise. To distinguish between both, the approximate inner area of the cycles is computed using the Shoelace formula (Equation (1)).

$$A = \frac{1}{2} \left| \sum_{i=1}^{n} (x_i \cdot y_{i+1} - y_i \cdot x_{i+1}) \right| \tag{1}$$

The orientation of each polygon is then calculated using Equation (2), so that the cycles can be reoriented clockwise, or counterclockwise, accordingly.

$$O = \sum_{i=1}^{n} (x_{i+1} - x_i) \cdot (y_{i+2} - y_i) - (y_{i+1} - y_i) \cdot (x_{i+2} - x_i) \tag{2}$$

9. *Semantic allocation.* The area covered by each connected component is spatially combined with semantic data from the built mask layer in order to add this information as an attribute to the polygon.
10. *Projection.* Finally, the vector data are reprojected, according to the geographic projection of the georeferenced map image, and exported in shapefile format.

This automatic pipeline delivers high-quality vector data, ensuring efficient and streamlined 3D modeling processes. Polygon adjacency is preserved, and neighboring polygons share the same nodes, facilitating manual correction if necessary. What is more, the level of geometric simplification of the vertices is easily parameterized, due to the $\epsilon$ parameter. This is made possible by focusing on the vectorization of delimiting segments, rather than the polygons themselves. Detecting corners using the Harris algorithm, and intersections, helps maintain sharp geometries.

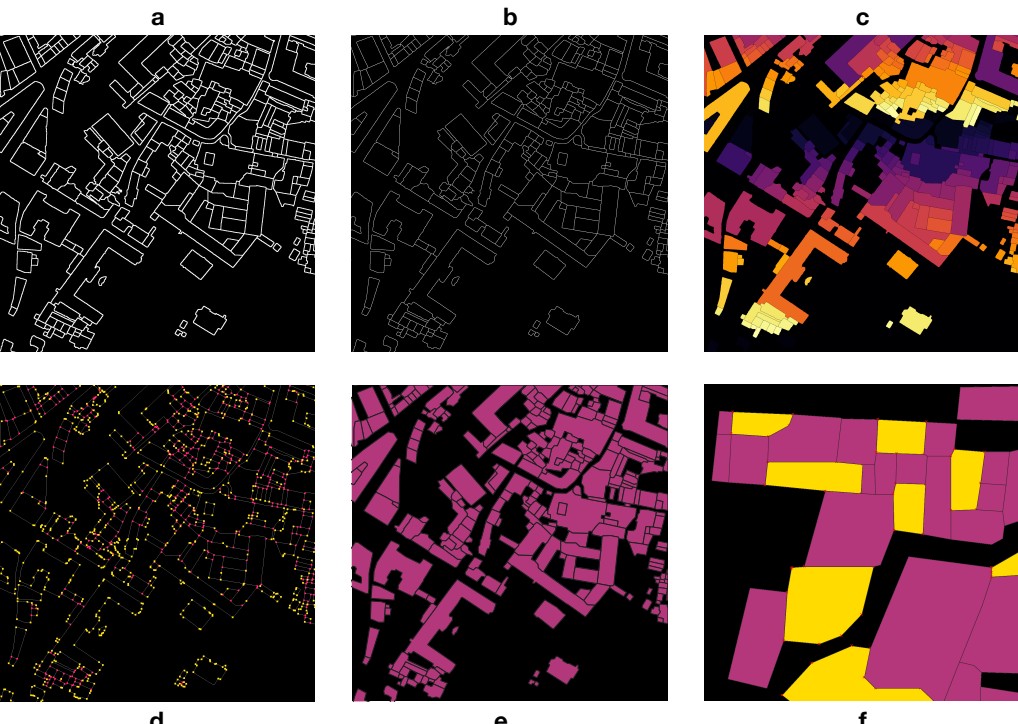

**Figure 1.** Illustration of the vectorization pipeline: (**a**) Output of the semantic segmentation of contours. (**b**) Thinned contours, step 1. (**c**) Connected components as color gradient, step 2. (**d**) Detection of corners (in yellow) and intersections (in red), steps 4 and 5. (**e**) Result of the vectorization (buildings in carmine). (**f**) Close-up of the resulting vector data in QGIS (some polygons, in yellow, are selected to show their points).

Subsequently, a spatial comparison between the 1940 dataset and the other vectors is performed to determine whether the presence of the buildings persists in time. This enables the creation of a diachronic spatial dataset. To account for minor misalignment, we apply a buffer around the polygons. However, it is important to note that this automated process does not detect changes in shape, but only captures the appearance and disappearance of polygons. For this specific case study, we considered the latter an acceptable trade-off, and intervened manually in case of macroscopic inaccuracy. The dataset is then further enriched by operating a spatial joint with the Atlas of Jerusalem [32], a secondary cartographic source from the 20th century, indicating the number of floors, the type of roofs, and other architectural features. These complementary data are stored as categorical attributes.

### 2.2. Parameters Employed for the Procedural Modeling

Once good-quality vector data have been obtained, we can prepare them for the procedural modeling process. As such, it is essential to encode the 2D GIS datasets in a format that can accommodate 3D information while preserving all available attributes and the geographic location of the features. For this task, CityJSON [23], a JSON encoding of CityGML, has been chosen due to its readability and simplicity. To adhere to this format, it is necessary to transfer the fields from our GIS files into a hierarchical tree structure, and ensure their compliance with the specified format. This adaptation allows for the inclusion of all the required fields essential for the procedural generation of 3D models. This extension builds upon our initial proposal [25], incorporating improvements and additional features to support the refined level of modeling proposed in this contribution. The extension ensures compatibility between the scripts and the parameterization.

To start the 3D procedural modeling, a minimum set of parameters is required. As we will see, they can be either provided by the user, or predicted automatically when necessary. We will now go over them, and subdivide them according to the two levels of detail.

For LOD1, a set of three parameters is used to define the height of the extrusion:

- Height ($H$);
- Floor Height ($H_f$);
- Number of Floors ($N_f$).

The relationship between these parameters is evident. The total height ($H$) of the building is calculated by multiplying the number of floors ($N_f$) by the floor height ($H_f$). This relationship ensures that the extrusion process accurately represents the specified number of floors, maintaining the proportional scaling of the building's vertical dimensions. This parameter interdependence becomes particularly valuable in situations where explicit height information is missing from the dataset, but the number of floors is known (e.g., in the case of a birds-eye-view data source). In situations where the height or number of floors is unknown, we employ a predictive method to estimate the number of floors and/or floor height, thereby enabling the calculation of the total height. By adopting this approach, we generate plausible height values, even in instances where the dataset lacks precise height information. With this approach, we generate credible height values, even when the dataset lacks specific height information.

For LOD2.1, the roof is added to the building model. The necessary parameters vary depending on the type of roof being modeled. For this contribution, we developed four possible roof types: *hip*, *gable*, *flat*, and *domed* roofs. The Table 1 provides an overview of the parameters needed for each one of them.

**Table 1.** Parameters needed for the procedural modeling of each type of roof.

| Hip | Gable | Flat | Domed |
|---|---|---|---|
| Base floor thickness | Base floor thickness | Base floor thickness | Base floor thickness |
| Slope | Slope | Railing height | Railing height [1] |
| Upper floor thickness | Upper floor thickness | Railing width | Railing width [1] |
| Eaves overhang | Eaves overhang | | Dome horizontal radius (%) |
| | | | Dome vertical radius (%) |

For *hip* and *gable* roofs, the "Base floor thickness" parameter represents the thickness of the roof's base floor. The "Slope" parameter indicates the angle or pitch of the roof expressed as a value between 0 and 1, while the "Upper floor thickness" parameter denotes the thickness of the upper floor of the roof, i.e., the vertical thickness of the sloped surfaces. The "Eaves overhang" parameter represents the horizontal extension of the roof beyond the exterior walls (i.e., the amount of horizontal offset).

For *flat* roofs, in addition to the "Base floor thickness", some parameters describe the railing or balustrade around the perimeter.

For *domed* roofs, additional parameters come into play to reproduce the typical lowered dome shape that can be found in Jerusalem. In particular, for each footprint, we compute the center and radius of the maximum circle inscribed in the concave polygon. This geometrical problem, also referred to as the "poles of inaccessibility" problem, can be solved employing an iterative grid algorithm [44] inspired by Garcia-Castellanos and Lombardo's algorithm [45]. Once we obtain the maximum circumference, we can define the value (expressed in $]0, 1]$ of its radius to use as the base circumference for the dome. Now, using the vertical value, we can decide whether the dome is a full dome or a lowered one. The chosen value, within the range $]0, 1]$, determines the height of the highest point of the dome, where 1 represents the full dome and 0 corresponds to a null height. Subsequently, we find the sphere passing through the base circumference and the point that we just obtained to determine the dome shape.

As exemplified, parameters enable us to define the key characteristics of each roof type, ensuring that the procedural modeling accurately represents the desired architectural

features. This empowers users to customize the roofs of their 3D models, and tailor the architectural style.

However, when working with datasets of cities in the past, it is actually uncommon to possess all the necessary information to perform procedural modeling. Data incompleteness, limitations of historical records, or the stage of the data collection can prevent researchers from proceeding. For this reason, we need to infer all missing values.

*2.3. Addressing the Issue of Missing Parameters*

To address the challenge of missing data, we developed a solution that takes into account two potential cases: (1) when a parameter (e.g., roof type) is completely missing from the dataset, or (2) when it is partially present, but incomplete.

In the case where a numerical parameter (e.g., the slope of a hip or gable roof) is completely missing, we simply sample the value from a statistical distribution, either informed, or based on a educated guess. The choice of the range inside which this value should be picked strongly depends on the specific case study, and its definition could be based—depending on the availability of information—on an educated guess, surveys and statistical evidence, local regulations, or primary and secondary sources. For instance, Ref. [8] reports previous works that used local regulations or architectural principles to derive the maximum height of buildings. By default, we assume that inside of this range, the missing numerical parameter is likely to follow a normal distribution centered around a parameterizable mean value. The choice of using a normal distribution is based on the ideal assumption that the missing parameter values form a Gaussian distribution around the mean value. However, in practice, the specific distribution pattern may vary depending on the context and nature of the parameters being considered. Further statistical tests on the normality of parameter distributions should be conducted in the future to validate this assumption, as 4D historical data become available for a larger number of cities and in a variety of cultural contexts. When culturally related datasets are available, we provide the ability to sample the values from the real distribution. For instance, Ref. [10] present the distribution of building height in the region of Hamburg. This distribution is, unsurprisingly, multimodal (bimodal in their case), as the building height is tightly connected to the number of storeys. If a project were to focus on another Hanseatic city, the distribution could prove better than a normal approximation. In this case, we provide the ability to fill missing values by performing a random choice with replacement from any provided distribution. When the missing parameter is categorical (e.g., the type of roof), the logic is similar. By default, the random choice is performed using uniform probability between all categories. However, we also let the user provide a custom probability for each category. In the case of Hamburg, for instance, flat and hip/gabled roofs account for 2/3 and 3/10, respectively, of the roofs. In the case of Jerusalem, the proportions are similar, with 3/5 and 1/3, respectively. However, domed roofs are predominant in Jerusalem. In this case, the user might want to personalize the probability associated with each category.

The second case we consider is when a parameter—either numerical or categorical—is partially present in the dataset. In this case, we propose leveraging the knowledge available for the other buildings in the city to infer the missing values. This approach has been successfully tested in previous work, such as the work by Farella et al. [15], where building heights were inferred using similar techniques. In our framework, we propose the use of a machine learning model, such as Random Forest, to infer any parameter (whether numerical or categorical) that is incompletely present in the dataset. This requires the selection of—among all available parameters—the explanatory variables, i.e., the parameters we consider relevant for predicting the target variable. For instance, variables that are assigned arbitrarily to buildings, such as the id code, need to be excluded from this process, since their value cannot be logically related to the actual features of the building.

In the Jerusalem case study, we focused on two target variables: the number of floors and the type of roof. Both parameters were only partially available. In Section 2.4,

we implement four different machine learning classifiers and compare their inference performance.

*2.4. Filling Gaps*

We tested and compared four classifiers implemented in scikit-learn [46] to select the most suitable one: Decision Tree, Random Forest [47], Adaptive Boosting [48], and Gradient Boosting [49]. Each classifier was trained to predict two different target features: the number of floors and the type of roof.

The selected models were chosen, among others, due to their ability to handle missing values in the input, to provide interpretable decisions, and due to their training speed. Indeed, values can be missing both in the input and in the output. While some approaches attempt to fill missing values in the input with mean values, or simply exclude incomplete rows, the naturally incomplete nature of historical data compels us to use algorithms that natively manage missing values. Moreover, the training speed and the energy efficiency of these models is crucial, as we may want to dynamically retrain the model when new information is added to the dataset.

One additional advantage of using Decision Tree, Random Forest, or Adaptive Boosting models is their interpretability. In fact, they allow us to visualize the decision-making process and understand how the model reaches its predictions. This transparency can be valuable in understanding the relationships between different features and the inferred parameter values. It also provides precious insight into the dataset, and helps validate the model's predictions.

To identify the better-performing model, a confusion matrix was computed for both roof type classification and number of floors. Additionally, we compared the score against a baseline, by measuring the accuracy obtained when assigning the most frequent categorical value.

Our evaluation was conducted on the presented test dataset of the city of Jerusalem between 1840 and 1940, with the aim of predicting the values for the number of floors and the type of roof. However, the data that the Atlas of Jerusalem provides on these two factors cannot be directly mapped into a single value without losing or flattening information. For instance, inside the Old City, the majority of buildings report a range of 1–3 floors, instead of a precise numerical value. However, for the purpose of encoding in the CityJSON format, we need to arbitrarily assign a single numerical value to use for procedural modeling. This does not constitute an issue for the creation of a 3D CityJSON, since we can still keep track of the nuances of the original information through paradata. However, we decided that these values should not be taken into consideration for training the machine learning model, as we would not want to introduce an unnecessary set of flattened data. In the same way, in the secondary sources of Jerusalem, tall buildings are encoded as "6+" outside the walls and "4+" inside the walls, which is rather imprecise. The information on the roof type presents a similar issue: hip and gable roofs are, respectively, described as "gabled (tiles)" and "slanted (other than tiles)". Even with the help of the book accompanying the Atlas of Jerusalem, it is not clear whether we can actually identify a specific corresponding distinction, or whether they denote the same shape, just characterized by a different construction technique or material. For this reason, for the purpose of evaluating the inference of values, we decided to consider them as a single category (hip/gable).

In order to evaluate the four machine learning models, we perform the training and inference directly on the values that were obtained from the Atlas, prior to the encoding process. In this way, we avoid imposing unnecessary simplification on the information that was extracted from secondary sources. However, the methodology that we present is presumably applicable to any geodata, and to any categorical field. For the training, we added to these columns three input features that were directly derived from the vector data: area of the polygon, and latitude and longitude of the object centroid. These additional features were incorporated to enhance the training process and improve the prediction

accuracy, by providing more context. To clarify the explanation, in Table 2, we present our dataset and the initial completeness of the various fields.

**Table 2.** Completion status of the main fields in our dataset.

| (Total) | Number of Floors | Roof Type | Material | Start Year | End Year |
|---|---|---|---|---|---|
| **Entries** (10,833) | 8405 | 8607 | 8563 | 10,817 | 10,817 |
| **Percentage** (100%) | 77.59% | 79.45% | 79.05% | 99.85% | 99.85% |

We evaluate the quality and speed of each of the selected models for predicting the two aforementioned parameters. In a real use case, we would only apply the classifiers to the instances that comprise missing values. However, for evaluation purposes, we will simulate this situation on a subset of the dataset for which the target is known. The value for the number of floors, for instance, is known for 8607 rows. For each experiment, we allocated 60% of this subset for training purposes, and the remaining 40% for validation and testing (see Table 3).

**Table 3.** Rows available for each of the two target parameters and number of rows resulting from the subdivision between training and test sets.

| | #Rows (Number of Floors) | #Rows (Roof Type) |
|---|---|---|
| **Total (100%)** | 8405 | 8607 |
| **Training set (60%)** | 5043 | 5164 |
| **Test set (40%)** | 3362 | 3443 |

As anticipated, we trained four different machine learning algorithms, testing various configurations for the most promising ones:

1.  **Random Forest** with (a) max_depth = 5 and n_estimators = 10, (b) max_depth = 5 and n_estimators = 200, (c) max_depth = 50 and n_estimators = 200;
2.  **Decision Tree** with (a) max_depth = 5, (b) max_depth = 50, max_depth = 100;
3.  **AdaBoost**;
4.  **Gradient Boosting Classifier**.

The first evaluation is based on calculating the respective accuracy of each method and plotting a confusion matrix to visualize the results of the predictions for each experiment (Figure 2). The score corresponds to the mean classification accuracy (Equation (1)).

$$\text{Accuracy} = \frac{\text{Number of Correct Predictions}}{\text{Total Number of Predictions}} \tag{3}$$

This result can be compared with the baseline accuracy, or the score that would be obtained by assigning to all rows the most frequently encountered value (naive solution to the problem). In Figure 2, we show the results obtained for the prediction of roof type and the number of floors for each category. As anticipated, in order to accommodate the data provided in the secondary source without losing information, we consider the number of floors as categorical data as well. This is made possible due to the flexibility of the selected models in handling both numerical and categorical information.

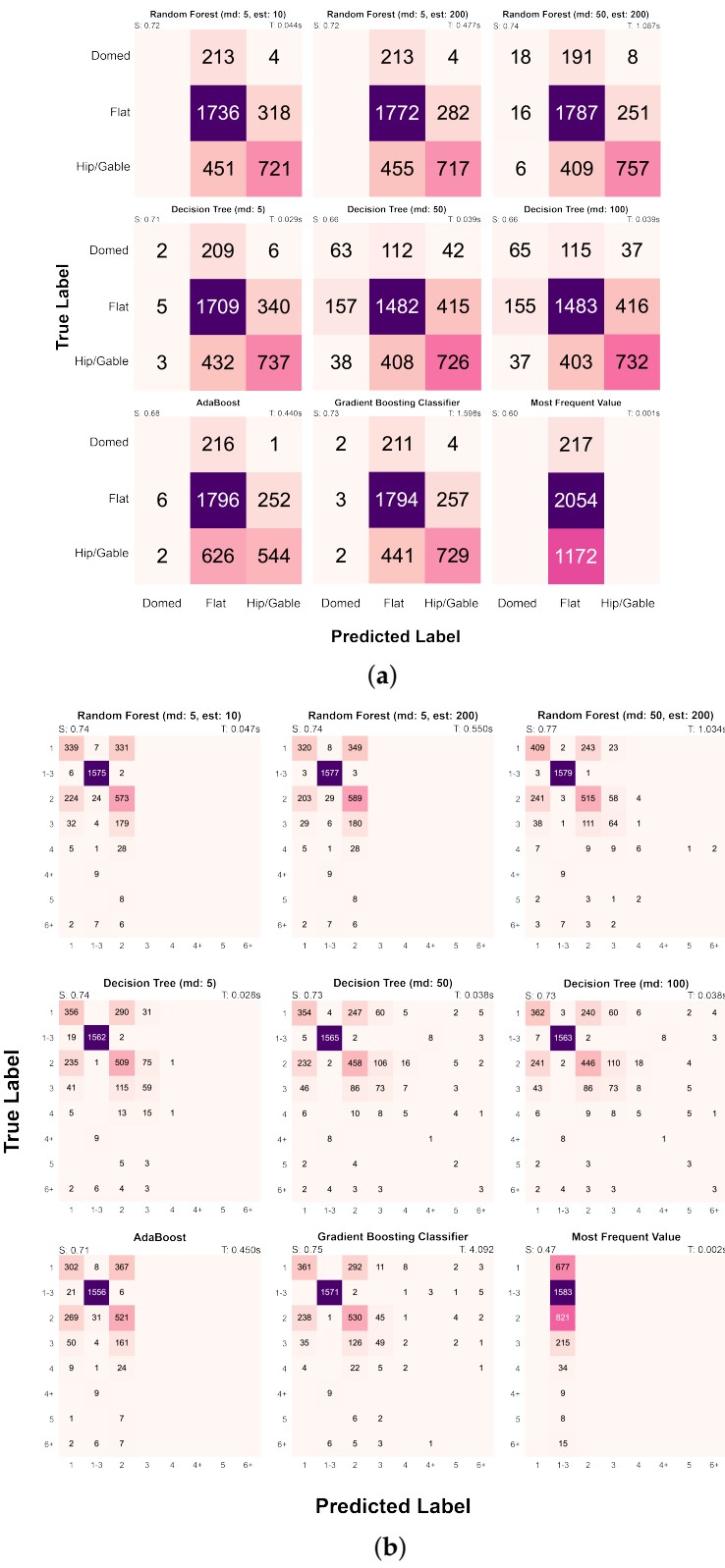

**Figure 2.** The plots are organized according to the target parameter: (**a**) On the top, results for the type of roof. (**b**) On the bottom, results for the number of floors. Each plot reports the confusion matrix, score, and computation time for each tested model. Rows represent the true value, while columns represent the predicted value. Values on the diagonal going from the top-left to the bottom-right corners are correctly predicted. The last confusion matrix in each of the two plots represents the result that we would obtain with the naive solution (baseline accuracy). A darker colour corresponds to a higher number of values.

As anticipated, we seek a solution that maximizes the score while minimizing the training time (Figure 3). This criteria allows us to quickly retrain the model whenever new information is added to the dataset. This enables us to obtain more accurate predictions in a timely manner. By optimizing both the performance and speed of the model, we can ensure its practical applicability and adaptability to evolving datasets. This perspective can also be considered a form of active learning, with this meaning a situation where the machine learning model is interactively fed with new training samples. A human expert working on the dataset could therefore progress more efficiently, by validating the predictions of the machine learning model that are true and correcting only faulty ones.

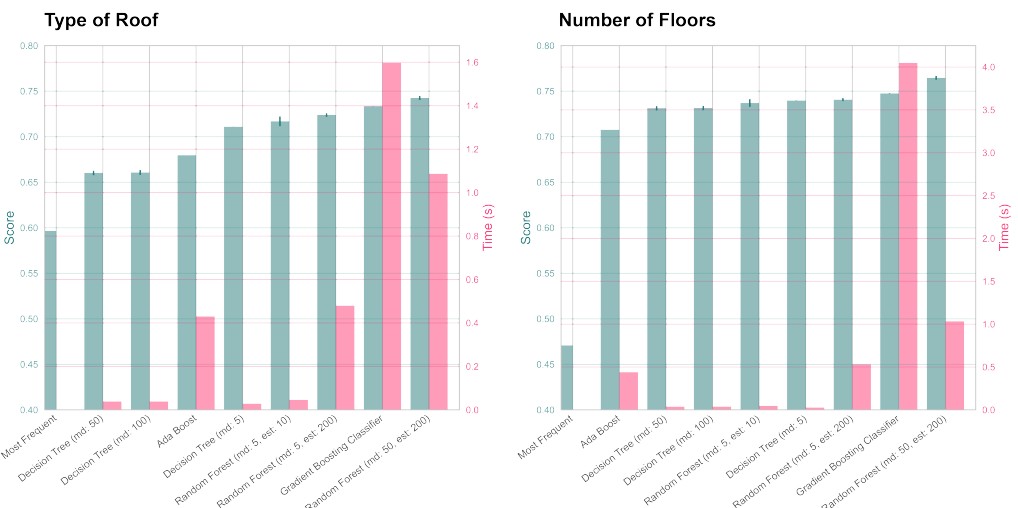

**Figure 3.** Accuracy (in light-blue) and calculation time (in pink), for each experiment (x-axis). The score is compared to the baseline accuracy. The lines at the top of the bars represent the standard deviation.

Considering the results, Random Forest appears to be the most adequate method. Moreover, considering the improvement shown with regard to the score obtained with the naive solution, the approach proves to be generally relevant. However, considering the highly specific nature of the problem, we considered it irrelevant to perform a quantitative comparison with results obtained in other studies. Instead, we decided to focus on comparing the performances of the various algorithms, and to understand their robustness to small datasets, as this can often be the case for historical case studies.

In fact, we also focused on addressing the limitations of this approach when dealing with extremely rarefied datasets for a specific value. For instance, in the case where a target field is rarely documented, we might have a very limited number of rows that could be used as a training set. By progressively decreasing the number of available data points, we aimed to determine the threshold at which a machine learning model becomes less effective than simply filling the missing rows with the most frequent value. In order to assess this, we randomly drew an increasingly large subset, beginning with 0.1% of the initial (corresponding to 5 rows over the 5043 rows available for the number of floors) up to the full dataset (100%). For each experiment, we retrained the machine learning classifiers, repeating the operation 200 times, with various random subsets, for each scale. We then computed the mean accuracy and the standard deviation for comparison.

As visible in Figure 4, even when few data are available, we rapidly exceed the baseline accuracy, and even reach a reasonable performance. In our particular case, the results already start stabilizing when we reach 1% of the dataset, or around 50 training examples, for the number of floors. A few more samples are necessary when predicting the type of roof.

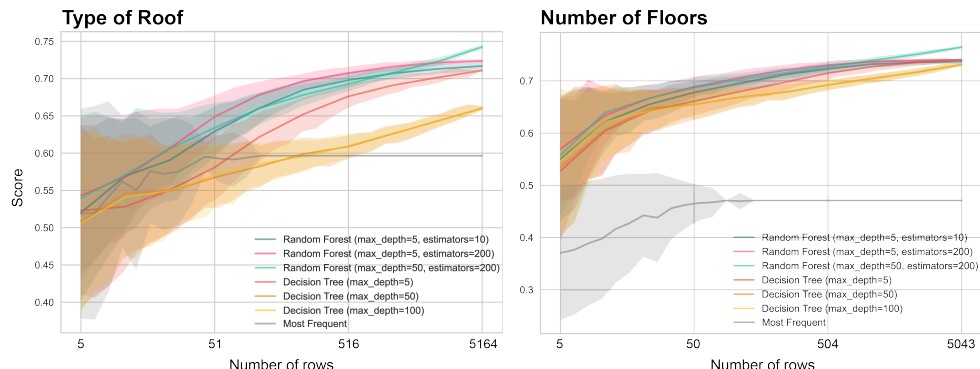

**Figure 4.** Mean classification accuracy and standard deviation for the various machine learning models, computed over 200 experiments. The x-axis shows the number of rows that were employed for training (in logarithmic scale). For each experiment, we picked *n* random samples from the full training dataset, and repeated the training process 200 times. Then, we computed the mean accuracy and standard deviation. We plot this against the naive solution to see the improvement provided by each method when the available training data are consistently reduced. The results obtained for the Adaptive Boosting model and the Gradient Boosting classifier are shown in Appendix A.

### 2.5. Transforming 2D Geodata into a 3D CityJSON Model with dhCityModeler

The core principle of the dhCityModeler module is based on the idea that every 3D geometry is the direct result of the combination between a footprint and a set of parameters. By leveraging this approach, whenever new information becomes available, it is sufficient to encode it inside our CityJSON: subsequently, the procedural modeling scripts can be relaunched, generating new 3D geometries that reflect the newly added information. This flexible framework allows for seamless integration of additional data, enabling the continuous refinement and enhancement of the 3D models.

Moreover, the available procedural modeling tools are generally able to produce models in which roofs are simplified as plain sloped surfaces, without incorporating features such as overhang and thickness (i.e., LOD 2.0). Although these models can provide a reasonable approximation of reality, more refined models that incorporate these features can significantly reduce deviations from the real world, and yield better results in spatial analyses [31]. For this reason, the scripts that we propose are able to generate the geometry with an improved LOD equal to 2.1.

In practice, the module that we propose works as a converter: it takes as an input a GIS file and converts it into a 3D CityJSON version of it. This process involves (1) mapping the available fields and geometries (LOD 0) to the Historical CityJSON extension; (2) automatically retrieving the terrain mesh for the area of interest and encoding it in CityJSON as a TINRelief; (3) filling the partially missing values (only the ones related to procedural modeling) by applying a Random Forest algorithm; (4) assigning values to the fields that are still required for modeling, but are completely absent from the dataset, by applying a normal distribution falling within a range of plausible values; (5) moving the footprints at the level of the terrain; and (6) proceeding with the procedural modeling and encoding of the LOD 1.0 and LOD 2.1 geometries.

We will now go over each of these operations, explaining in detail how they work.

The initial step consists of mapping the tabular-form attributes of our GIS file to the hierarchical structure of CityJSON, according to the fields proposed in the Historical CityJSON extension. Before proceeding with this step, it is necessary for the user to provide the geodata with the fields readily prepared to be correctly mapped. For instance, in our case, we first proceeded with converting the values obtained from the secondary source in the correct format: the categories that we had for the number of floors had to be converted to a numerical value, while recording the original value in the comments of the paradata. In particular, where the number of floors was indicated as "1–3", we randomly picked a value between three, assigning a slightly higher probability for values equal to 1 or 2

($p$ = 0.35). Similarly, where, for the type of roof, a value could correspond either to a hip or gable roof, we decided to randomly assign one of them.

In this preliminary step, the geometry is simply encoded as an LOD 0 Multisurface positioned at a null height. As a result of this preparatory step, we achieve a straightforward and direct conversion from a shapefile or GeoJSON format to a CityJSON file.

The second preparatory step involves integrating the terrain mesh into the CityJSON file and adjusting the elevation of the building footprints. The footprints, now encoded as Multisurfaces, need, in fact, to be translated vertically to match the elevation data.

Initially, we calculate the geographic bounding box for our dataset, and use the Google Elevation API or OpenTopoData to retrieve elevation data. These grid elevation data are then used to generate a Triangulated Irregular Network (TIN) using the Pydelatin python library [50]. The delatin algorithm, based on the paper by Garl and Heckbert [51], is employed for approximating a height field using Delaunay triangulation. The resulting mesh is then encoded in the CityJSON format as a TINRelief, following the prescribed standard.

Subsequently, we look for the vertical value at which every footprint should be positioned. In order to do so, for each footprint, we calculate the elevation of the projection of the vertices onto the terrain mesh. By employing Delaunay triangulation, we can locate the triangle containing the projected point, and then interpolate the elevations of the triangle's vertices to obtain the sought value. The smallest elevation, among the ones obtained for each vertex, is employed to then translate the footprint vertically. The new LOD0 geometries are saved in CityJSON, now positioned at the correct height with respect to the terrain information. The next step involves inferring the missing values for CityJSON, and then proceeding with the procedural generation of LOD1 and LOD2 geometries.

By applying the methodology described in Section 2.4, we are able to infer partially missing parameters. In particular, if a parameter that is required for the successive phase of procedural modeling is partially missing, we use a Random Forest algorithm to complete the dataset. When a value is completely missing from the dataset, however, we simply select it using a normal distribution in a plausible range (when the value is numerical), or by randomly selecting it from a list of options (when categorical), with the possibility of personalizing the probabilities for each category, as anticipated.

For the purpose of our research, of the many typologies of procedural modeling, we decided to employ Constructive Solid Geometry (CSG). The CSG approach is based on the successive application of Boolean operators on simple solid shapes to obtain more complex ones. In order to do so, we employed CadQuery [52], a Python module designed for building parametric 3D CAD models with the capabilities of dealing with BREP geometries and performing spatial queries on geometry to ease the modeling procedures.

The generation of LOD1 simply consists of extruding the footprints of a given quantity. In particular, as presented in Section 2.2, we derive height to use for extrusion by combining the number of floors and the floor height, depending on the information available in CityJSON.

The creation of the LOD 2.1 model consists of adding the roof shape to the previously generated LOD 1 model. So far, our module can deal with the generation of four types of roofs: (1) *hip* **roofs**, that is, a roof that presents sloped surfaces on every side; (2) *gable* **roofs**, i.e., a roof that presents at least one vertical side (constituted by a portion of a wall); (3) *flat* **roofs**, which may or may not have a balustrade around their perimeter; and (4) *domed* **roofs**, i.e., flat roofs that also have a lowered dome on their surface (a category that was added to take into account the specificities of the case study).

To illustrate this point, we will briefly present the algorithms concerned with the generation of hip and gable roofs. The generation of hip roofs in particular is directly connected to the straight skeleton identification [53]. In our case, we directly employ 3D Boolean operations to compute the straight skeleton, proposing a volumetric approach to the solution of the problem that also works in the presence of concave angles. In Figure 5, we illustrate graphically the steps of our algorithm applied to each vertex iteratively. The algorithm is based on subtractions from a central shape. To retrieve the shape to subtract

each time, we calculate the offset distance that is necessary to avoid self-intersections that would cause errors when computing the 3D lofts.

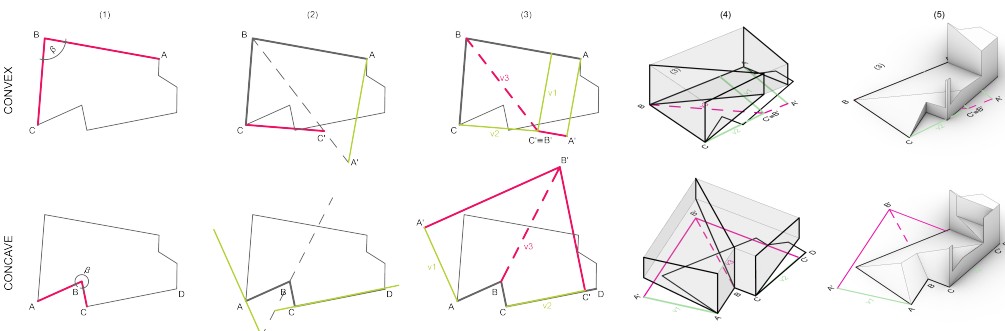

**Figure 5.** Illustration of the process followed to create the subtraction shape corresponding to each vertex. On the top row, we illustrate what happens when the angle in B is convex, while on the bottom row, the process follows the situation where the angle is concave. The iterative process, using B as the middle point, follows these steps: (1) We select the three consecutive points (A, B, C) and the side segments joining them. (2) For each side segment (AB and BC), we calculate the length of the segments from A and C that are perpendicular to the side segments, and intersect the bisector passing in B (AA', CC'). This length can be calculated by multiplying the length of the considered side for the tangent of $\beta/2$, where $\beta$ is the angle in B ($AA' = AB * tan(\beta/2)$). We employ the minimum between the two lengths as the offset distance. Note that when the angle is concave, we employ an arbitrarily large distance M, since self-intersection is not an issue in this case. (3) By applying the obtained length as the module of $v_1$ and $v_2$ (vectors that are perpendicular to AB and BC), we retrieve the length of the bisector in B that should be used as an offset vector for the sides. However, we use the vectors $v_1$ and $v_2$ to trim this offset, and finally find the shape of our subtraction. (4) The subtraction volume is obtained through a loft operation. (5) The subtraction volume is finally employed against the initial extrusion of the roof base shape with a Boolean subtraction.

Gable roofs are generated starting from the correspondent hip roof, as shown in Figure 6. Given a polygonal base shape, we identify the sloped sides that present only three vertices. This is particularly easy with CadQuery, due to the available geometric filtering capabilities that make it possible to select only the faces that are sloped and present only three points, with the additional condition of having two points on the polygon's boundary (this constraint is necessary to avoid corner cases where a triangular face is not on the perimeter of the polygon). Then, we construct the volumes that are needed to fill the roofs, and we perform a Boolean union operation.

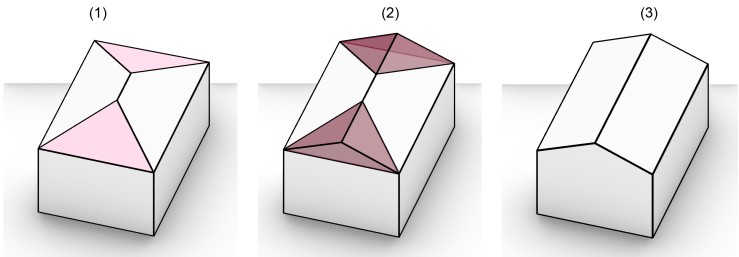

**Figure 6.** Illustration of the process of creation of the base shape for gable roofs. (1) First, we detect the faces that are positioned along the perimeter, and present three vertices (highlighted in pink). (2) Then, we create the solid shapes required to fill the gable roof, by finding its vertices and then creating the faces joining them. (3) We apply a Boolean union and obtain the desired shape.

The result of this process corresponds to a simple set of sloped faces (i.e., LOD 2.0), but we actually want to obtain a more realistic representation of the roof, with overhangs and thickness (LOD 2.1). To this end, we leverage the querying operations of Cadquery to select

the faces that we need at each step and create a shell, which we apply on top of our base roof shapes (either hip or gable). In Figure 7, we illustrate the methodology employed to generate the shell. In particular, the process entails the following steps: (1) We take the base polygonal shape of the roof and apply an horizontal offset equal to the selected overhang. (2) We generate the corresponding hip or gable roof shape using the previously described functions. (3) We query the ridge lines of the base roof and the one we just generated and compute their vertical distance. We filter the sloped faces of the second roof and translate them vertically back at the height of the base roof, using the quantity that we just found. (4) We extrude the faces vertically of a given roof thickness.

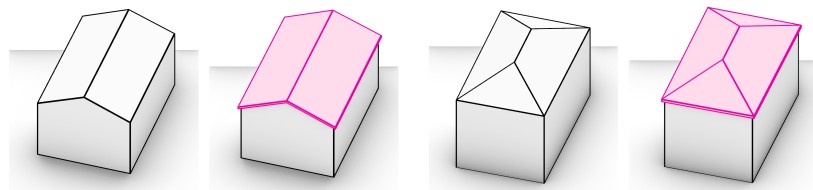

**Figure 7.** Illustration of the generation of the shell for gable and hip roofs. By adding the shell (highlighted in pink), we push the LOD of our models from LOD 2.0 to LOD 2.1.

### 3. Resulting Model

In this study, we performed a process of data completion of a 2D dataset and successfully transformed it into a 3D CityJSON representation. The tools that were presented make our pipeline robust to missing data. In this way, the production of a high-LOD model is not hindered by common incompleteness in the dataset. Our approach thus makes visualization possible, while providing a simple tool to convert a 2D dataset into a 3D format that supports a hierarchical structure for encoding additional information. In particular, in Figure 8, we show the detailed view of the model that is obtained using the proposed procedural modeling scripts. The level of detail (LOD 2.1) in the 3D representation of roof shapes we achieve is satisfactory.

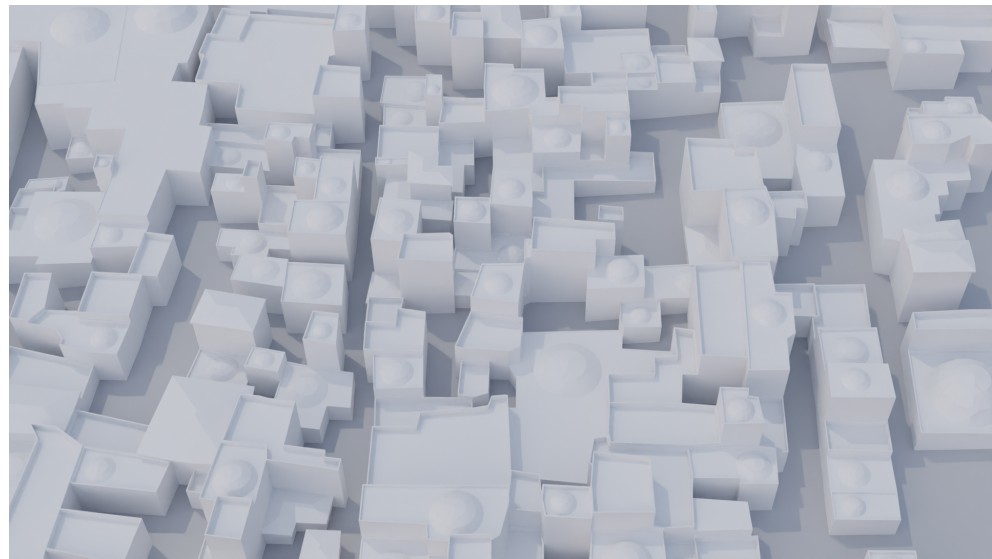

**Figure 8.** Detailed view of the LOD 2.1 model resulting from the application of the described methodology over every building of the dataset. Image rendered using Blender.

The resulting model is also not limited to geometric information. While the final result of traditional 3D modeling consists of a geometric model, the methodology we propose is designed to create an informative model. In fact, the advantage of combining the 3D format with a hierarchical data structure lies in the possibility of keeping track not only of the parameters that were used, but also of their origin, and the assumptions or even hypotheses that were made. In particular, by employing the CityJSON extension we proposed in [25], we are able to keep track of the provenance of each attribute, using the CityJSON hierarchical structure (Figure 9).

Thus, the outcome of our work consists of an informative model that maintains all the initial attributes mapped from the original geodata.

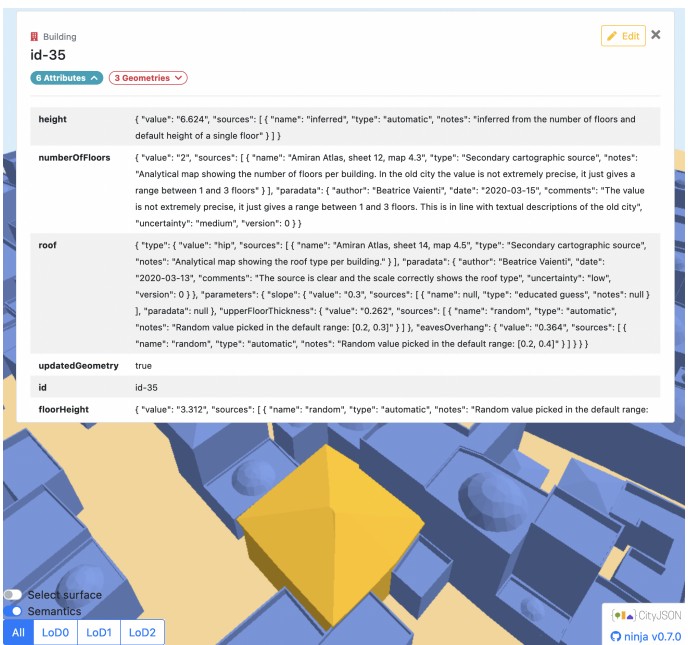

**Figure 9.** In the image, we can see part of the attributes that characterize a selected building (in yellow). The visualization was performed using the CityJSON webviewer Ninja [54].

The use of the CityJSON hierarchical structure makes it possible to embed the diachronic information inside the model and extract the desired time model from the full dataset (Figure 10).

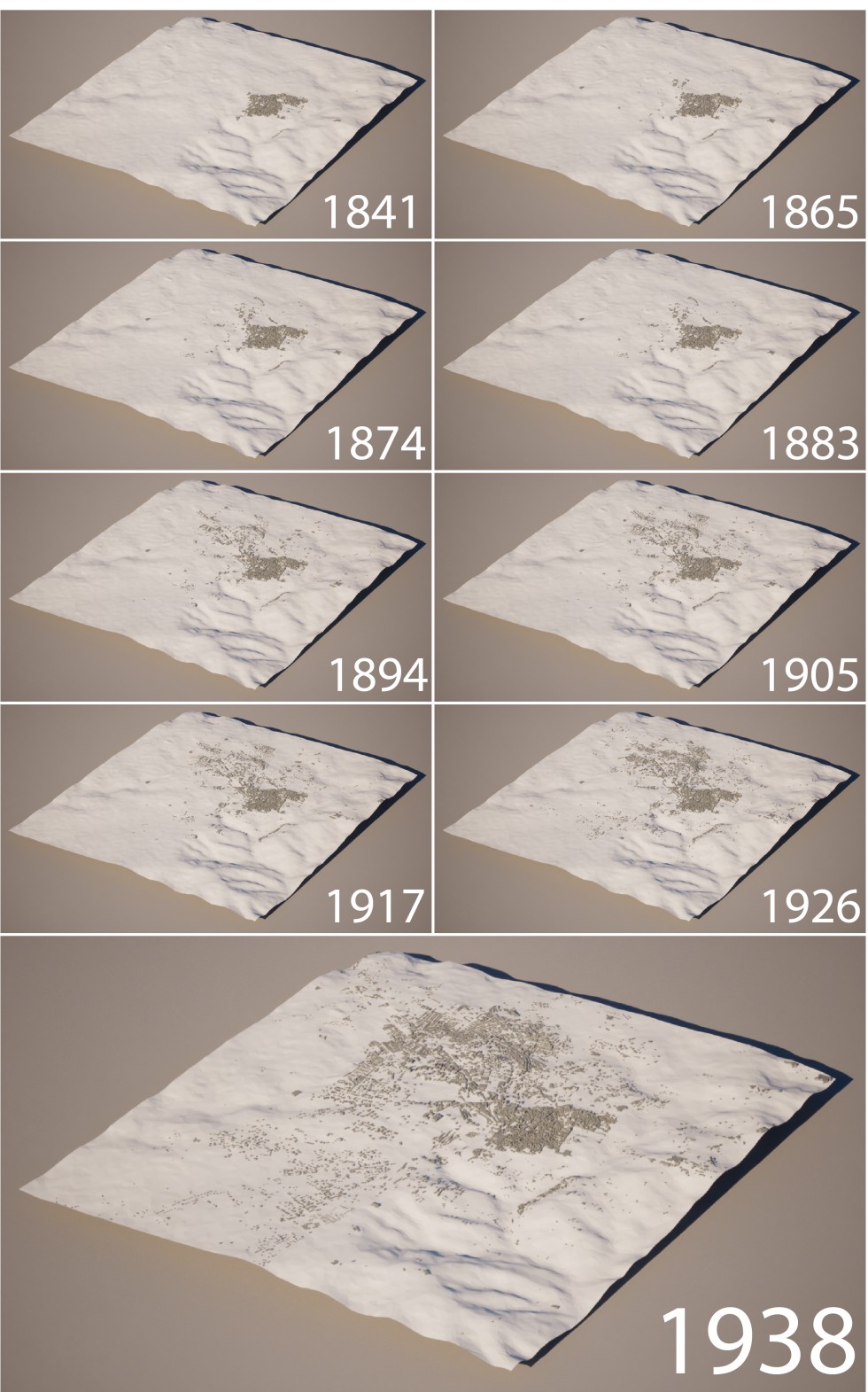

**Figure 10.** Complete view of nine temporal phases corresponding to our 4D model of the city of Jerusalem.

## 4. Discussion

The conversion from a 2D GIS file to a 3D CityJSON involved several preparatory steps, including mapping the attributes to the CityJSON tree structure and the incorporation of a terrain mesh from Google API or OpenTopoData. This provides the user with a simple tool

that makes it possible to transform any 2D geolocated dataset to a 3D geolocated CityJSON that incorporates three different levels of detail. Our procedural modeling scripts addressed the current limitations of tools that offer this kind of service, improving the representation of roofs with thickness and overhang. Moreover, the generated 3D models incorporate detailed information about building footprints, parameters, and their sources, ensuring traceability, and therefore, transparency.

In addition to the technical tools that were developed, our first contribution consists of proving the advantages of employing computationally efficient machine learning models for data completion even in the case of small datasets and with categorical data. We selected Random Forest as the algorithm that best suited our case, both with respect to speed and accuracy. The latter confirms previous results obtained by Farella et al. for the prediction of building height (and thus, a numerical value) [15]. A great advantage of using Random Forest lies in the possibility of training the model even in the presence of missing input values, as well as in the possibility of visualizing the decision trees that make up the Random Forest classifier.

Indeed, we consider other advantages of deploying machine learning methods. The fact that, from a limited number of available fields, it is still possible to fill up a dataset automatically in a significant way could mean that machine learning could help leveraging some of the hidden patterns that connect attributes in the sparse dataset. Furthermore, as users, we can actively refine the results by adding new information when available, or by confirming the result of an inference by applying an educated guess. Moreover, compared to previous studies, our tools offer the advantage of being generic, enabling the user to select the target feature along with the desired predictors.

For this reason, one of the main advantages of the proposed approach lies in its adaptability to the growth of the dataset. When dealing with historical data, in fact, we often have to cross-examine different sources, and new pieces of information may become available in time. However, traditional methods for 3D reconstruction, especially if dealing with manual modeling, do not present the flexibility to incorporate new information. In the proposed approach, not only do we present a system that is capable of actively learning, but we also handle the procedural modeling as a result of the information present in the dataset and the footprints. As a consequence, the 3D geometry only corresponds to a temporary geometric representation of the data, rather than an alleged final output. Due to these two elements, the model can grow incrementally without risking any data loss, while bootstrapping the reconstructive process. The possibility of using versioning [55] in CityJSON files further enhances the idea of 3D reconstructions as ever-growing models, rather than final outputs, ultimately opening the way to collaborative usage.

## 5. Conclusions

In this article, we have tackled the challenges for the generation of informative 3D or 4D urban historical reconstructions. Our methodology was specifically designed to address the recurring issues related to the creation of such models: data incompleteness, cultural specificity, the iterative nature of scientific projects, and the subjectivity of reconstruction and interpretation. By facing each of these issues, we present a comprehensive framework that introduces a set of open-source tools for transforming 2D GIS datasets into 3D CityJSON representations automatically.

We showcase the operations implemented to generate the initial 2D geodata from a collection of digitized historical map images. This demonstration underscores the versatility of the suggested pipeline, showcasing its potential for application in other comparable historical case studies. This process consisted of the semantic segmentation of the digitized maps and their vectorization with an algorithm especially designed to produce simplified yet sharp and continuous vector geometries, adapted to procedural modeling.

Machine learning techniques have been integrated into our framework to allow for intelligent completion of missing values in the dataset, with the aim of facing the issue of data incompleteness. Among the various methods, Random Forest achieved the best

results for the prediction of categorical information. Moreover, these algorithms present additional advantages: they are inherently resilient to empty fields in the input data, and they provide transparent and interpretable models. The visualization of decision trees, for instance, could provide valuable insight for the theory of architecture. Moreover, these light, low-resource models can be trained in real time, paving the way for active learning approaches and interactive interfaces.

The completed dataset was then structured in the CityJSON format, from which we generated the 3D model of the city with multiple LODs (0.0, 1.0, 2.1). The resulting models provided a more accurate and detailed representation of the built environment, enabling better spatial analyses and simulations. The CityJSON extension that we employed, which records the sources and paradata for each parameter, ensures traceability and enhances data provenance, thus tackling effectively the challenge of subjectivity of interpretation by making the reconstructive process transparent. The transparency and accessibility are also ensured by making the tools open-source. Moreover, by providing a procedural modeling library that is customizable by the user, we managed to address the cultural specificity that is inherent in every historical reconstruction, creating a tool that can be adapted to each case study.

The machine learning approaches we propose are deliberately simple, which allows for dynamic and iterative retraining based on a few training samples. By combining procedural geometry generation, we effectively meet the requirement of constructing dynamic models that can iteratively adapt to newly acquired information. This approach ensures the flexibility and responsiveness necessary to accommodate evolving data and maintain the accuracy and relevance of the models. It is true, however, that larger and more complex datasets could benefit from more advanced and computationally expensive algorithms; to remedy this, we envision that such models could be retrained periodically, or one could perform bootstrapping to create models for larger regional projects, while simpler models could focus on spreading the changes to city or neighborhood scales. Other future work includes incorporating the tools presented here into a web platform to take full advantage of the dynamic capabilities of our framework.

In conclusion, our paper illustrates the successful transformation of 2D datasets into 3D CityJSON representations by incorporating machine learning techniques and advanced LOD modeling. This study contributes to the advancement of generic 3D modeling techniques that are very useful for urban planning and the architectural restitution of complex sets of buildings, and their application is especially suitable for the historical reconstruction of cities.

**Author Contributions:** Conceptualization, B.V. and R.P.; methodology, B.V. and R.P.; code, B.V. and R.P.; validation, B.V.; writing—original draft preparation, B.V. and R.P.; writing—review and editing, I.d.L. and F.K.; supervision, I.d.L. and F.K.; project administration, I.d.L. and F.K.; funding acquisition, I.d.L. and F.K. All authors have read and agreed to the published version of the manuscript.

**Funding:** This project has received funding from the European Union's Horizon 2020 research and innovation programme under the Marie Skłodowska-Curie grant agreement No 945363, and the College of Humanities at EPFL

**Data Availability Statement:** The tools that were presented are available at the following URL: https://github.com/BeatriceVaienti/dhCityModeler.

**Conflicts of Interest:** The authors declare no conflicts of interest.

## Abbreviations

The following abbreviations are used in this manuscript:

LOD    Level Of Detail
GIS    Geographic Information Systems

**Appendix A**

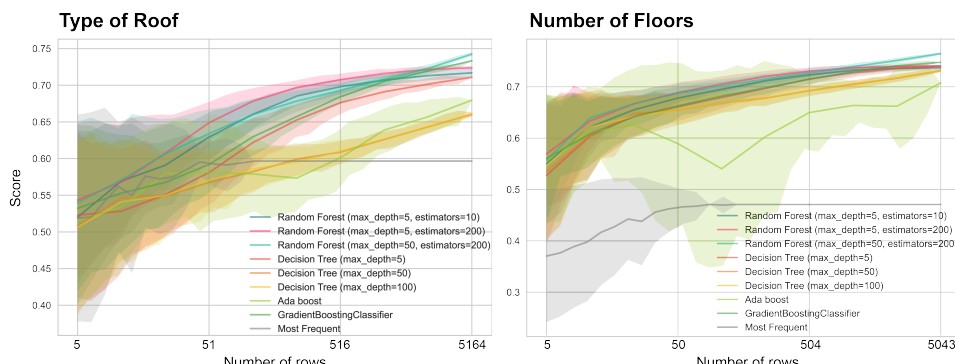

**Figure A1.** Mean classification accuracy and standard deviation for the various machine learning models, computed over 200 experiments. The x-axis shows the number of rows that were employed for the training process (in logarithmic scale). For each experiment, we picked *n* random samples from the full training dataset, and repeated the training process 200 times. Then, we computed the mean and standard deviation of the obtained score. We plot this against the naive solution to see the improvements provided by each methods even when the available training data are consistently reduced. In this plot, the Ada Boost model and the Gradient Boosting classifier are also shown. While the Gradient Boosting classifier exhibits good performances, but was then excluded due to computational time, the Ada Boost model shows less reliable performances, even when adding data to the training set. One can in fact notice a high standard deviation in the scores, with less stable results when compared with the other predictors.

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
