# Peer review of "Machine-Learning-Enhanced Procedural Modeling for 4D Historical Cities Reconstruction"

_remotesensing, doi:10.3390/rs15133352_

Round 1
Reviewer 1 Report
The research topic is very interesting and is described pertinently, with punctual bibliographic references and with excellent English. All the steps, from the vectorisation of historical city maps to the elaboration of 3D models, are clear and easy to understand. However, for greater completeness, I would suggest accompanying the text with an explanatory final image, which portrays the city chosen as a case study in three dimensions, namely Jerusalem, in its evolutionary path.
Publication is recommended.
Author Response
We would like to express our gratitude to the reviewer for their valuable suggestion: we incorporated a new image in Section 3, to showcase the city’s evolution in nine moments in time (Figure 10).
Reviewer 2 Report
This paper developed an open-source Python module that fills gaps in 2D GIS datasets and directly generates 3D models up to LOD 2.1 from GIS files, allowing for effective management of spatial data and the generation of detailed 3D models. The topic is valuable, but several points could be improved:
1. The introduction includes many general statements that are common knowledge in the GIS community. However, the authors need to effectively connect this background information to the paper's main purpose, which is Procedural Modeling for 4D Historical Cities Reconstruction. It is unclear what specific problem they are addressing, and there needs to be more explanation of why they chose this particular methodology. Moreover, this section could be improved by making it more concise and objective, with a more precise connection to the paper's focus; there seem to be more than 2000 words in this section.
2. In the introduction, the authors should present the rationale behind their choice of the Machine Learning Enhanced idea and explain why specific references are mentioned while others of equal importance are omitted. For example, deep-learning approaches. Additionally, the authors should describe how they implemented the Machine Learning-Enhanced idea in their investigation and how it relates to the overall methodology of the study.
3. When describing the proposed method, it is essential to highlight the key innovation of your work and how it compares to related works. While lines 183-187 may indicate your motivation, the innovation is unclear. To improve the clarity, it may be necessary to specifically elaborate on how the proposed method differs from existing methods and how it adds value to this field.
4. When evaluating the proposed method, it is important to compare it with the current state-of-the-art and demonstrate its innovation. However, the authors only presented the results without comparing them with existing methods. To strengthen the validity and significance of the proposed method, it is recommended to provide a comparison with state-of-the-art techniques.
5. In the Discussion section, it is important to summarize the evidence for each conclusion and state your findings clearly. For instance, in lines 634-635, where it was stated that "our first contribution consists of proving the advantages of employing light machine learning models," the definition of "light" was not clear. To make these conclusions more precise and objective, it is recommended to provide more details about the specific definition of "light" in this context and how it was evaluated. Additionally, presenting the principles shown by the results more clearly and comprehensively is suggested.
6. The Conclusion section has too many repeats of the earlier section. I suggested providing a concise and clear summary of the significant findings from the study, along with their implications.
There are many expression errors, indicating that the paper needs to be carefully checked for these errors.
Author Response
We would like to thank the reviewer for their feedback. We have considered their suggestions and made significant revisions based on their comments:
1. The introduction includes many general statements that are common knowledge in the GIS community. However, the authors need to effectively connect this background information to the paper's main purpose, which is Procedural Modeling for 4D Historical Cities Reconstruction. It is unclear what specific problem they are addressing, and there needs to be more explanation of why they chose this particular methodology. Moreover, this section could be improved by making it more concise and objective, with a more precise connection to the paper's focus; there seem to be more than 2000 words in this section.
In response to this we have made significant revisions to the introduction. Firstly, we have removed the part that was generally discussing the advantages of using GIS, as too broad and general. We then just focused on presenting experiences that were using GIS as the basis for procedural modeling [lines 140-143].
Then, to improve the clarity of the choices behind our methodology and the objectivity of the introduction we also chose to organise it to follow four main challenges associated with the creation of urban historical reconstructions (i.e. data incompleteness [lines 36-66], cultural specificity [lines 67-79], the iterative nature of scientific projects [lines 80-99], and the subjectivity of reconstruction and interpretation [lines 100-132]). Each challenge is now presented in relation to our decision-making process for developing our framework, providing a clearer explanation of the specific problem we are addressing and the reasons behind our chosen methodology. In order to make this section more concise we also decided to move the technical part that was describing the type of procedural modeling in Section 2.5 [lines 589-594].
2. In the introduction, the authors should present the rationale behind their choice of the Machine Learning Enhanced idea and explain why specific references are mentioned while others of equal importance are omitted. For example, deep-learning approaches. Additionally, the authors should describe how they implemented the Machine Learning-Enhanced idea in their investigation and how it relates to the overall methodology of the study.
We revised the part of the introduction that introduces the machine learning approaches to data completion and added several new references [lines 48-60]. We have presented a reference to show an example of deep-learning approaches applied to remote sensing data and a survey that shows a trend towards classical machine learning algorithms to deal with simpler datasets such as already vectorised geodata. Then, we presented seven examples that use these algorithms to fill GIS contemporary datasets to show the potential of these approaches for this specific task.
We better explained the rationale behind the Machine Learning-Enhanced idea by connecting it with the four challenges followed in the introduction (and listed in the previous answer). In this way we motivated this approach on one hand to deal with data incompleteness and on the other to deal with the need of dynamicity and customisation. We also decided to recall these two points in the conclusion [lines 713-714 and 732-735].
3. When describing the proposed method, it is essential to highlight the key innovation of your work and how it compares to related works. While lines 183-187 may indicate your motivation, the innovation is unclear. To improve the clarity, it may be necessary to specifically elaborate on how the proposed method differs from existing methods and how it adds value to this field.
In Section 2, we furtherly elaborated on the added value of the tools that we propose.
E.g. in the part corresponding to vectorization:
“The step of vectorizing 2D geometries is a demanding part of the 3D model creation process. On the one hand, geometries must be simplified to avoid the aliasing effect inherited from the raster output of the neural network. Secondly, it must avoid the pitfalls of existing vectorization algorithms, such as OpenCV and scikit-image's contour functions, or QGIS's raster-to-vector module. These out of the box tools often compute geometries independently for each polygon. This leads to inconsistent results, such as aliasing, neighboring polygons not sharing edges, and undesirable slivers and overlaps. To the contrary, desirable qualities of a vectorization algorithm designed for 3D data generation would include keeping sharp building corners and the ability to parameterize the level of simplification of the vertices, without affecting the local coherence.”
“The completed dataset was then structured in the CityJSON format, from which we generated the 3D model of the city with multiple LODs (0.0, 1.0, 2.1). The resulting models provided a more accurate and detailed representation of the built environment, enabling better spatial analyses and simulations. The CityJSON extension that we employed, which records the sources and paradata for each parameter, ensures traceability and enhances data provenance, thus tackling effectively the challenge of subjectivity of interpretation by making the reconstructive process transparent. The transparency and accessibility are also ensured by making the tools open-source. Moreover, by providing a procedural modeling library that is customizable by the user we managed to address the cultural specificity that is inherent to every historical reconstruction, creating a tool that can be adapted to each case study.”
4. When evaluating the proposed method, it is important to compare it with the current state-of-the-art and demonstrate its innovation. However, the authors only presented the results without comparing them with existing methods. To strengthen the validity and significance of the proposed method, it is recommended to provide a comparison with state-of-the-art techniques.
We emphasized the originality and novelty of the proposed method by improving the discussion and conclusion sections. We highlighted the key achievements of our tools by emphasizing that they constitute a highly open and customizable solution [lines 682-684, 707-709, 728-730] that joins together a series of techniques to solve recurring challenges in the historical reconstructions domain. In the Conclusion, we decided to follow the narrative structure of the revised introduction and use the selected four main challenges as a way of explaining the novelty and significance of our proposal [lines 700-705]:
“Our methodology was specifically designed to address the recurring issues related to the creation of such models: data incompleteness, cultural specificity, the iterative nature of scientific projects, and the subjectivity of reconstruction and interpretation. By facing each of these issues, we presented a comprehensive framework that introduces a set of open-source tools for transforming 2D GIS datasets into 3D CityJSON representations automatically.”
Regarding quantitative comparison with state-of-the-art techniques, at the beginning of Section 2 [lines 195-197] and in Section 2.4 [lines 502-506] we decided to explain that for what regards the results of the machine learning algorithms in filling the dataset, a direct quantitative comparison with other case studies would be not very relevant, because of the extremely high dependency of the results from the employed dataset [lines 502-506]. Since our aim is to provide a customizable tool, we were more interested in an internal comparison between the models regarding performances, accuracy, and robustness to small datasets.
5. In the Discussion section, it is important to summarize the evidence for each conclusion and state your findings clearly. For instance, in lines 634-635, where it was stated that "our first contribution consists of proving the advantages of employing light machine learning models," the definition of "light" was not clear. To make these conclusions more precise and objective, it is recommended to provide more details about the specific definition of "light" in this context and how it was evaluated. Additionally, presenting the principles shown by the results more clearly and comprehensively is suggested.
We rephrased the Discussion section and added other explanations to clarify. We avoided addressing the chosen models as “light” and instead addressed them as “computationally efficient” in order to be more clear.
"In addition to the technical tools that were developed, our first contribution consists in proving the advantages of employing computationally efficient machine learning models for data completion even in the case of small datasets and with categorical data. We selected Random Forest as the algorithm that best suited our case, both with respect to speed and accuracy. The latter confirms previous results obtained by Farella et al. for the prediction of the buildings' height (and thus a numerical value) \cite{farella2021}. A great advantage of using Random Forest lies in the possibility of training the model even in presence of missing input values, as well as in the possibility of visualizing the decision trees that make up the random forest classifier."
6. The Conclusion section has too many repeats of the earlier section. I suggested providing a concise and clear summary of the significant findings from the study, along with their implications.
The entire Conclusion section underwent significant change to be more clear on the findings and implications of our methodology. In particular, by following the same new structure of the introduction, and thus the four envisioned challenges, we tried to clarify how the proposed tools answer these issues.
Reviewer 3 Report
Summary/Contribution: This work developed a framework for historical city reconstruction that uses procedural modeling and machine learning models in a Geographic Information System (GIS) framework to build detailed 3D reconstructions of former cities. An open-source Python module in the framework fills gaps in 2D GIS datasets and builds LOD 2.1 3D models from GIS files. CityJSON supports historical models and ensures interoperability. The practical case study employing footprints of the Old City of Jerusalem between 1840 and 1940 shows the dataset's development, completion, and 3D representation, demonstrating the approach's versatility and usefulness. This project creates instructive 3D historical city models to improve accessibility and accuracy. Procedural modeling eliminates the need for additional software and simplifies historical city reconstruction.
Comments/Suggestions:
1. A shorter, more precise thesis statement that expresses the paper's goal and contribution would be beneficial for the introduction.
2. The authors might include more particular examples of the uses of historical city 3D models for academic research and teaching, as well as the advantages of employing 3D models for population projections and visibility evaluations.
3. To enhance transparency and reproducibility in the modeling process, the authors should go into more detail on how important it is to describe metadata and paradata.
4. The authors could give more particular instances of the difficulties in implementing CityGML and how CityJSON resolves these issues.
6. The specific algorithms and methods utilized for the semantic segmentation of the historical maps and the vectorization of building footprints may use a more thorough description in section 2. This would make it easier for readers to comprehend the approach's technical aspects and how it compares to currently used techniques and tools.
7. The difficulties and constraints of the machine learning approach for completing gaps in the dataset, as well as the potential biases and uncertainties induced by this procedure, may be better illustrated by the authors using more precise examples.
8. The parameters that are either entirely or partially absent from the dataset and how they affect the precision and fidelity of the final 3D model may be better illustrated by the authors.
9. The statistical approaches used to estimate missing numerical parameters, and how they connect to existing methods and tools, may use a more thorough discussion in section 2.3. The range of values utilized for the normal distribution and the methodology used to determine them might both be clarified by the authors.
10. The authors might include more specific instances of categorical parameters that are entirely absent from the dataset and explain how this affects the precision and fidelity of the final 3D model. Additionally, they may explain how the user can customize the probability distribution that is used to choose values at random and how it works.
11. The authors might include more detailed examples of the uses of the suggested technique and the resulting model in urban planning, cultural preservation, or other fields. This would make it easier for readers to comprehend the approach's useful implications and potential advantages.
12. The CityJSON format's hierarchical data structure and how it permits the encoding of additional information may use a more thorough description in section 3. The authors could go into more detail on the precise traits and parameters that were used to create the model as well as how those factors related to the original geodata.
13. The potential biases and uncertainties generated by the use of machine learning algorithms for filling in missing data, and how they might be addressed or managed, could be covered in greater detail in section 4. The authors may also go into greater detail on how accuracy and performance might be traded off in various scenarios and how those tradeoffs might be optimized.
14. Section 4 could use a more thorough explanation of the potential drawbacks and difficulties of the suggested approach, such as the requirement for manual building footprint annotation, potential errors brought on by the use of outside data sources, or potential biases brought on by the choice of machine learning algorithms and parameters.
16. The suggested approach might be compared to current techniques and tools for 3D reconstruction and visualization of urban landscapes, and the authors could give more specific examples of how it could improve or supplement those techniques and tools. They could also include further details on the suggested approach's potential effects and relevance for the larger research community and society.
18. The authors might want to take into account the following sources that are pertinent to this subject:
a.https://ieeexplore.ieee.org/abstract/document/9842406
b. https://incose.onlinelibrary.wiley.com/doi/abs/10.1002/inst.12434
Can be improved
Author Response
Thank you for the feedback provided. Below, we outline our responses and the corresponding actions we have taken:
1. A shorter, more precise thesis statement that expresses the paper's goal and contribution would be beneficial for the introduction
We added a short sentence at the beginning of the Introduction to anticipate the aim of the paper [lines 32-35].
2. The authors might include more particular examples of the uses of historical city 3D models for academic research and teaching, as well as the advantages of employing 3D models for population projections and visibility evaluations.
We added two examples showing on one hand how to employ 3D urban models to estimate the population applying different techniques, and another one presenting a volumetric approach to visibility evaluation using 3D models [lines 27-28].
3. To enhance transparency and reproducibility in the modeling process, the authors should go into more detail on how important it is to describe metadata and paradata.
We decided to better organize the challenges of our framework and isolated the challenge of “subjectivity of reconstruction and interpretation” to better explain why we need metadata and paradata [lines 100-112].
4. The authors could give more particular instances of the difficulties in implementing CityGML and how CityJSON resolves these issues.
To better explain the advantages of using CityJSON we employed the term developer-friendly (instead of user-friendly) and compact instead of lightweight [lines 119-120]. In fact, as stated in the cited article (ref. 23 in the paper), CityJSON files are on average 6 times more compact than cityGML files, and it is built with developers in mind, thanks to a variety of open tools such as the Ninja web viewer. The ease of creating an extension is also a major element in favor of CityJSON, as discussed in the introduction [lines 122-125]. We also refer to our previous work (ref. 25 in the paper) that deals also with that particular question.
5. The choice of the particular tools and approaches used in the research, as well as how they contribute to the paper's overarching purpose, might have been explained more fully by the authors.
We considerably updated the introduction, the method and conclusion to improve the explanations on our both standpoint and the chosen methodology. We improved the motivation of our approaches by revising the structure of the introduction: now we present our solutions by following point by point four main challenges that we have to face, in order to create a historical 3D urban reconstruction (i.e. data incompleteness [lines 36-66], cultural specificity [lines 67-79], the iterative nature of scientific projects [lines 80-99], and the subjectivity of reconstruction and interpretation [lines 100-132])
6. The specific algorithms and methods utilized for the semantic segmentation of the historical maps and the vectorization of building footprints may use a more thorough description in section 2. This would make it easier for readers to comprehend the approach's technical aspects and how it compares to currently used techniques and tools.
We updated the text in section 2.1. We attempted to make the explanations and the formulation clearer throughout the text. We also enriched the description of the vectorization pipeline, more specifically the seventh step:
"To reconstruct the polygons, the nodes directly adjacent to each connected component are retrieved. The segments are adjacent to the polygon if both their starting and their terminal node is adjacent to the connected component. Then, simply following the adjacent segments one after the other makes it possible to reconstruct the cycle of each polygon."
We rewrote the eight step, introducing two new equations to make the maths behind the algorithm less abstract :
"Polygons can encompass "donut holes", which should be oriented counterclockwise, according to the shapefile format convention, while the outer cycle should be oriented clockwise. To distinguish between both, the approximate inner area of the cycles is computed using the Shoelace formula (Eq. 1).
[Equation 1]
The orientation of each polygon is then calculated using Eq. 2, so that the cycles can be reoriented clockwise, or counterclockwise, accordingly.
[Equation 2]."
7. The difficulties and constraints of the machine learning approach for completing gaps in the dataset, as well as the potential biases and uncertainties induced by this procedure, may be better illustrated by the authors using more precise examples.
We thank the reviewer for their suggestion. We updated the paper at several places to address this point.
8. The parameters that are either entirely or partially absent from the dataset and how they affect the precision and fidelity of the final 3D model may be better illustrated by the authors.
As they are completed using the machine learning approach, the parameters are not missing anymore at the stage of generating the 3D model. Without a roof category parameter for instance, a LOD2 model would be unachievable.
9. The statistical approaches used to estimate missing numerical parameters, and how they connect to existing methods and tools, may use a more thorough discussion in section 2.3. The range of values utilized for the normal distribution and the methodology used to determine them might both be clarified by the authors.
We have provided a more thorough discussion in section 2.3 [lines 372-389]. We provided suggestions on how to determine the range boundaries and the distributions based on data availability and we provided an example of distribution for the height of buildings, as reported in a reference, to better explain the rationale behind our statistical approach.
10. The authors might include more specific instances of categorical parameters that are entirely absent from the dataset and explain how this affects the precision and fidelity of the final 3D model. Additionally, they may explain how the user can customize the probability distribution that is used to choose values at random and how it works.
The fidelity of the model is a very qualitative appreciation. We added some examples specifying the distribution of the roof types for Hamburg and Jerusalem, with the respective roof repartition. Depending on the choice of the user, it would be possible to decide on a probability repartition close to those study cases (e.g. â…” flat roofs, 3/10 hip or gabled) based on the similarity of the case under consideration with the reported cities. Alternatively, it would be possible to settle for a uniform distribution, with the risk of ending with a skewed reconstruction of the city.
11. The authors might include more detailed examples of the uses of the suggested technique and the resulting model in urban planning, cultural preservation, or other fields. This would make it easier for readers to comprehend the approach's useful implications and potential advantages.
We integrated two additional references in the initial part of the introduction that showcase the possibility of using these models for quantitative and qualitative urban analyses [lines 27-28].
12. The CityJSON format's hierarchical data structure and how it permits the encoding of additional information may use a more thorough description in section 3. The authors could go into more detail on the precise traits and parameters that were used to create the model as well as how those factors related to the original geodata.
Since going into detail in the structure prescribed for CityJSON files would demand a consistent amount of space, we considered enough to refer to the official reference for CityJSON format, which explains in detail all the characteristics of the encoding and our article describing the features added to the main encoding. Moreover we will release the extension file along with the described tools.
13. The potential biases and uncertainties generated by the use of machine learning algorithms for filling in missing data, and how they might be addressed or managed, could be covered in greater detail in section 4. The authors may also go into greater detail on how accuracy and performance might be traded off in various scenarios and how those tradeoffs might be optimized.
In our approach, we value transparency in the modeling process. Indeed, we keep track of every step, including when filling missing data. Therefore, we consider that uncertainties and biases are dealt with thanks to their exposure. Moreover, in the presented experiments, Random Forest proved both the most accurate and one of the fastest models. Therefore, the decision of picking it did not require a large trade-off.
14. Section 4 could use a more thorough explanation of the potential drawbacks and difficulties of the suggested approach, such as the requirement for manual building footprint annotation, potential errors brought on by the use of outside data sources, or potential biases brought on by the choice of machine learning algorithms and parameters.
The manual vectorisation of each map for a city the scale of Jerusalem could require several weeks of work for a trained specialist. For the 10 maps contained in our dataset, this could correspond to 6 months or more. Therefore, the annotation of a few patches from those maps represents a clear improvement.
Regarding the use of outside data sources, namely the Historical City Maps Semantic Segmentation Dataset, the latter was specifically developed to achieve Generic map segmentation, as mentioned in the related paper.
15. The conclusion could use a more in-depth discussion of the potential future directions and enhancements of the suggested methodology and produced model, such as the inclusion of more complex machine learning models, the integration with collaborative efforts, or the creation of interactive expert interfaces.
We thank the reviewer for their suggestion. We updated the conclusion accordingly :
"The machine learning approaches we propose are voluntarily simple, which enables dynamic, iterative retraining, on the base of few training samples already. This, combined with the procedural generation of the geometry successfully tackles the need for creating dynamic models, able to iteratively adapt to newly available information. However, larger and more complex dataset could benefit from more advanced and computationally expensive algorithms. Such models could be retrained periodically, or bootstrap the creation of models for larger regional projects, while simpler models could focus on diffusing modifications at the scale of the city or the neighborhood. Other future works involve embedding the presented tools into a web platform in order to fully benefit from the dynamic capabilities of our framework."
16. The suggested approach might be compared to current techniques and tools for 3D reconstruction and visualization of urban landscapes, and the authors could give more specific examples of how it could improve or supplement those techniques and tools. They could also include further details on the suggested approach's potential effects and relevance for the larger research community and society.
To better explain the potential of our framework in relation to previous examples and available tools we highlighted in the text its adherence to the following points:
- The tools are open source
- They are connected to the introduced four specific challenges of historical urban reconstruction
- Improved LOD
17. To increase the quality and significance of this paper, I suggest including a section on formal methods for AI-based technique verification. Formal methods, which use mathematical models and logic to verify system correctness, are becoming more and more important in the development and validation of AI-based techniques.
18. The authors might want to take into account the following sources that are pertinent to this subject:
- https://ieeexplore.ieee.org/abstract/document/9842406
- https://incose.onlinelibrary.wiley.com/doi/abs/10.1002/inst.12434
While the suggested line of inquiry provides material for valuable research, the extensive analysis required to address the reviewer’s comment would require substantial additional work that lies beyond the originally intended scope of the paper.
Round 2
Reviewer 2 Report
The authors have addressed my concerns.
Reviewer 3 Report
The authors considered my comments and suggestions. Good luck.
May be improved